# CERTIFIED ROBUSTNESS FOR DEEP EQUILIBRIUM MODELS VIA INTERVAL BOUND PROPAGATION

**Colin Wei**
Stanford University
colinwei@stanford.edu

**J. Zico Kolter**
CMU and Bosch Center for AI
zkolter@cs.cmu.edu

## ABSTRACT

Deep equilibrium layers (DEQs) have demonstrated promising performance and are competitive with standard explicit models on many benchmarks. However, little is known about certifying robustness for these models. Inspired by interval bound propagation (IBP), we propose the IBP-MonDEQ layer, a DEQ layer whose robustness can be verified by computing upper and lower interval bounds on the output. Our key insights are that these interval bounds can be obtained as the fixed-point solution to an IBP-inspired equilibrium equation, and furthermore, that this solution always exists and is unique when the layer obeys a certain parameterization. This fixed point can be interpreted as the result of applying IBP to an infinitely deep, weight-tied neural network, which may be of independent interest, as IBP bounds are typically unstable for deeper networks. Our empirical comparison reveals that models with IBP-MonDEQ layers can achieve comparable $\ell_\infty$ certified robustness to similarly-sized fully explicit networks.[1]

## 1 INTRODUCTION

A recent development in neural network design has been the introduction of *implicit* layers (Amos & Kolter, 2017; Chen et al., 2018; Agrawal et al., 2019; Bai et al., 2019; 2020; El Ghaoui et al., 2021), where the output is defined implicity as the solution to certain sets of conditions, rather than explicitly, via closed-form functions. These layers are promising alternatives to standard explicit deep learning layers and have demonstrated improved expressivity and inductive biases in a variety of settings, for example, processing time series (Rubanova et al., 2019), generative modeling (Grathwohl et al., 2018), solving logical reasoning tasks (Wang et al., 2019), solving two player games (Ling et al., 2018), and many others. One particularly promising class of implicit layers is deep equilibrium layers (DEQs) (Bai et al., 2019), which define the output as the solution to an input-dependent fixed point equation. DEQ-based models have matched or outperformed traditional explicit models even in commonly benchmarked settings (Bai et al., 2019; 2020).

Though recent empirical successes of DEQs have been promising, their implicit nature and inherent mathematical complexity also give rise to basic concerns. In order for DEQs to realize their promise, these concerns should ideally be mitigated or resolved. For example, one major issue with DEQs is well-posedness – a solution to the fixed point equation defining the layer might not exist. On the other hand, explicit layers always have well-defined outputs. A recent line of work has focused on addressing this important concern (Winston & Kolter, 2020; Revay et al., 2020; Xie et al., 2021).

This paper tackles a less-studied, but also important, question for DEQs: certified adversarial robustness. Because robustness is a basic concern for safe deployment of deep models (Szegedy et al., 2013; Goodfellow et al., 2014), for explicit models there is a large literature dedicated to certifying robustness, or guaranteeing correctness of the predictions even when the input is subject to imperceptible adversarial perturbations (see e.g. (Raghunathan et al., 2018a; Wong & Kolter, 2018; Gowal et al., 2018; Dvijotham et al.; Xiao et al., 2018; Cohen et al., 2019)). Many certified robustness methods require opening up the black box of the model and therefore only work for explicit models. It is unclear how to certify robustness of DEQs, which are only defined implicitly.

---

[1]Code is available here: https://github.com/cwein3/ibp-mondeq-code.

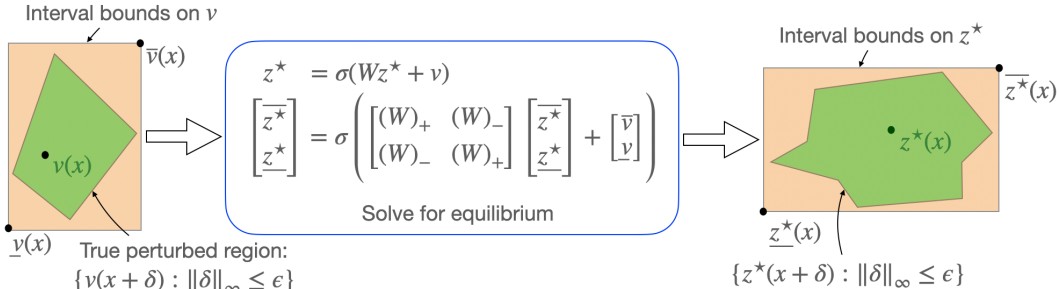

Figure 1: **Illustration of method.** We propose an equilibrium equation which takes as input $v(x)$ and interval bounds $\overline{v}(x), \underline{v}(x)$, and solves jointly for $z^\star$, the standard output of the DEQ, as well as $\overline{z^\star}, \underline{z^\star}$, interval upper and lower bounds on $z^\star$.

The certified robustness method motivating this work is interval bound propagation (IBP) (Mirman et al., 2018; Gowal et al., 2018), a simple and cheap way to certify robustness to $\ell_\infty$ perturbations. IBP computes layerwise upper and lower bounds for each coordinate of the adversarially perturbed hidden layers. These bounds follow basic rules of interval arithmetic and are simple to obtain in closed form as explicit functions of a layer's weights. However, it is unclear how to apply this idea when layers are only defined implicitly.

In this paper, we propose the IBP-MonDEQ, a deep equilibrium layer which is certifiably robust to $\ell_\infty$ perturbations. Motivated by principles from IBP, we define the IBP-MonDEQ output as the solution of an augmented fixed-point equation involving 3 quantities: the unperturbed output of the layer and upper and lower interval bounds on this output. As with IBP for explicit models, interval bounds on the IBP-MonDEQ output are computed during the forward pass of the network and can be composed with other layers to certify robustness of the entire model.

More concretely, we build upon monotone operator deep equilibrium (MonDEQ) layers proposed by Winston & Kolter (2020), which take the preceding layer $v(x)$ as input and outputs the solution $z^\star$, which is guaranteed existence and uniqueness, to the following fixed-point equation:

$$z^\star = \sigma(Wz^\star + v(x)) \tag{1.1}$$

We propose an IBP-inspired fixed-point equation

$$\begin{bmatrix} \overline{z^\star} \\ \underline{z^\star} \end{bmatrix} = \sigma\left( \begin{bmatrix} (W)_+ & (W)_- \\ (W)_- & (W)_+ \end{bmatrix} \begin{bmatrix} \overline{z^\star} \\ \underline{z^\star} \end{bmatrix} + \begin{bmatrix} \overline{v}(x) \\ \underline{v}(x) \end{bmatrix} \right) \tag{1.2}$$

which maps upper and lower interval bounds $\overline{v}, \underline{v}$ on $v$ to $\overline{z^\star}$ and $\underline{z^\star}$, which provide upper and lower interval bounds on $z^\star$. Figure 1 depicts this process. The augmented fixed-point equation is derived by unrolling the computation of $z^\star$ into an infinitely deep, explicit network and applying IBP to the forward pass of this network.

One immediate challenge is that it is not clear that a fixed point solution to (1.2) should always exist, especially given its interpretation as the result of applying IBP to an infinitely deep network. Indeed, a major drawback of IBP is that its performance degrades with deeper networks, as observed by Shi et al. (2021) and also shown in Figure 2 (left). One potential explanation for this failure is that IBP bounds tend to be unstable with depth and can diverge for deeper models (Figure 2, right).

On the other hand, we show that a unique fixed-point solution to (1.2) is guaranteed to exist when $W$ admits a simple unconstrained parameterization which is easy to enforce throughout training. Thus, our results pinpoint a class of infinitely deep networks for which IBP bounds are provably stable, which may be of independent interest.

We experimentally compare the proposed IBP-MonDEQ layer against IBP for standard explicit models. We consider common certified robustness benchmark settings and evaluate architectures of various sizes. Our results show that models with IBP-MonDEQ layers can achieve comparable $\ell_\infty$ certified robustness relative to fully explicit models with similar parameter counts.

In summary, our contributions are as follows: 1) We study the certified robustness of DEQs, proposing the IBP-MonDEQ, a class of DEQs with a guaranteed unique fixed point with provable interval bounds on the fixed point value. 2) The proposed IBP-MonDEQs form an expressive class of

infinitely-deep models for which IBP is provably stable, which may be of independent interest. 3) Our experiments demonstrate that IBP-MonDEQ layers are competitive with standard explicit layers for $\ell_\infty$-certified robustness.

## 2 BACKGROUND

$\ell_\infty$ **certified robustness.** Consider a $K$-way classification task with neural network classifier $F$ : $\mathbb{R}^d \to \mathbb{R}^K$. For a given input $x$ with true label $y$, the classifier $F$ is adversarially robust to $\ell_\infty$ perturbations with radius $\epsilon$ if

$$\min_{\delta \in \mathbb{R}^d : \|\delta\|_\infty \leqslant \epsilon} F(x + \delta)_y - F(x + \delta)_{y'} > 0 \ \forall y' \neq y \tag{2.1}$$

Safety-critical applications require certifying the robustness of $F$, i.e., verifying whether 2.1 holds. Directly optimizing over $\delta$ to verify (2.1) is challenging because the objective is non-convex (Madry et al., 2017). Thus, recent work on certified robustness has focused on verifying (2.1) via computationally tractable relaxations (Raghunathan et al., 2018a; Wong & Kolter, 2018; Dvijotham et al.; Raghunathan et al., 2018b; Weng et al., 2018; Gowal et al., 2018; Salman et al., 2019b).

**Interval bound propagation.** IBP is a computationally efficient method for certifying $\ell_\infty$ robustness of neural networks (Mirman et al., 2018; Gowal et al., 2018). It proposes to verify (2.1) via a (potentially loose) upper bound on $F(x + \delta)_y - F(x + \delta)_{y'}$ which is obtained by computing propagating upper and lower bounds on each layer through the forward pass of the network.

More precisely, let $z(x)$ compute some hidden layer of the network on input $x$. We say that $\bar{z}(x, \epsilon)$, $\underline{z}(x, \epsilon)$ are interval bounds on $z$ at $x$ for perturbation $\epsilon$ if the following holds for all coordinates $i$:

$$(\underline{z}(x, \epsilon))_i \leqslant \min_{\delta \in \mathbb{R}^d : \|\delta\|_\infty \leqslant \epsilon} (z(x + \delta))_i \leqslant \max_{\delta \in \mathbb{R}^d : \|\delta\|_\infty \leqslant \epsilon} (z(x + \delta))_i \leqslant (\bar{z}(x, \epsilon))_i \tag{2.2}$$

We omit dependencies on $x$ and $\epsilon$ when clear from context. Letting $k$ denote the layer index, the bounds $\bar{z}_k$ and $\underline{z}_k$ are obtained inductively via simple interval arithmetic. For an affine layer $z_k = W z_{k-1} + b$ and pre-computed bounds $\underline{z}_{k-1}, \bar{z}_{k-1}$, IBP computes $\bar{z}_k, \underline{z}_k$ as follows:

$$\begin{bmatrix} \bar{z}_k \\ \underline{z}_k \end{bmatrix} = \begin{bmatrix} (W)_+ & (W)_- \\ (W)_- & (W)_+ \end{bmatrix} \begin{bmatrix} \bar{z}_{k-1} \\ \underline{z}_{k-1} \end{bmatrix} + \begin{bmatrix} b \\ b \end{bmatrix} \tag{2.3}$$

Here $(W)_+ \triangleq \max\{W, 0\}$ and $(W)_- \triangleq \min\{W, 0\}$ denote the matrix $W$ with negative or positive values truncated to 0. For simplicity we focus on ReLU networks, with $\sigma$ denoting the ReLU activation. For layers $z_k = \sigma(z_{k-1})$ which apply $\sigma$ coordinate-wise, IBP computes $\bar{z}_k = \sigma(\bar{z}_{k-1})$ and $\underline{z}_k = \sigma(\underline{z}_{k-1})$. Initial bounds are obtained via $(\bar{z}_0(x, \epsilon), \underline{z}_0(x, \epsilon)) = (x + \epsilon \mathbf{1}, x - \epsilon \mathbf{1})$, where $\mathbf{1}$ denotes the all 1's vector.

The interval bounds are propagated through all layers of the network by following the simple rules above. To certify (2.1) for the whole network, one straightforward method is to confirm that the margins of the interval bounds on the logits are positive: $(\underline{F}(x, \epsilon))_y - (\bar{F}(x, \epsilon))_{y'} > 0 \ \forall y' \neq y$. One important note about IBP is that the bounds should be optimized during training in order for the method to provide nontrivial robustness guarantees.

**Monotone operator equilibrium networks.** Proposed by Winston & Kolter (2020), MonDEQs are a class of DEQs inspired by monotone operator theory (Ryu & Boyd, 2016) which have guaranteed unique fixed point solutions to the following equilibrium equation:

$$z^\star = \sigma(W z^\star + v(x)) \tag{2.4}$$

Let $I_{h \times h}$ denote the identity matrix on $h$ dimensions, with the subscript omitted when clear. A unique fixed-point solution is guaranteed for (2.4) for the following class of $W$:

**Proposition 2.1** ((Winston & Kolter, 2020)). *Suppose $W \in \mathbb{R}^{h \times h}$ satisfies that $I - W$ is positive definite (PD), i.e. $I - W > 0$. Then $\forall v(x) \in \mathbb{R}^h$, a solution $z^\star$ to (2.4) exists and is unique.*

Here $\mathbf{0}$ denotes the all 0's matrix and $A > \mathbf{0}$ indicates that $u^\top A u > 0$ for all nonzero $u$ (note that $A$ does not need to be symmetric). Winston & Kolter (2020) guarantee that $I - W$ is PD using the following unconstrained parameterization, which is enforced throughout training: $W = (1 - m)I - AA^\top + B - B^\top$, for positive hyperparameter $m$. Section 3.1 builds on these results to derive certified upper and lower bounds on $z^\star$.

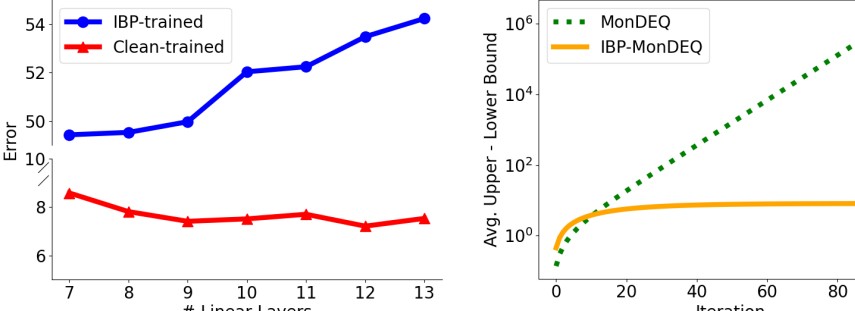

Figure 2: **Difficulties of IBP with depth. Left: error v.s. depth for IBP and standard training.** We train deeper models with IBP on CIFAR10 with $\epsilon = \frac{2}{255}$. We find that final certified error increases with the depth of the model. On the other hand, depth does not hurt when models are trained and evaluated on standard clean error. This suggests that IBP performs particularly poorly with depth. **Right: convergence of interval bounds for standard MonDEQs and our IBP-MonDEQ.** We plot the the average over coordinates of $\bar{z}_k - \underline{z}_k$, where $\bar{z}_k, \underline{z}_k$ denote bounds computed by the $k$-th iteration of the equilibrium solver for (3.3). This quantity blows up for the standard MonDEQ (Winston & Kolter, 2020) as the solver fails to find a fixed point. In contrast, the solver converges with our IBP-MonDEQ parameterization. Implementation details are in Section B.

## 3    CERTIFYING ROBUSTNESS OF MONDEQS USING IBP

In this section, we describe our core methodology for developing certifiably robust MonDEQ layers. We will first demonstrate how to obtain interval bounds for MonDEQ layers by computing the solution to a certain IBP-inspired fixed point equation (3.3). In Section 3.1, we characterize a new parameterization for $W$ for which a unique fixed point exists. In Section 3.3, we provide theoretical justification that the resulting IBP-MonDEQ layers remain expressive.

Our aim is to derive upper and lower interval bounds $\bar{z}^\star$ and $\underline{z}^\star$ for the fixed point solution to (2.4). A common interpretation of DEQs is that they compute an infinitely deep, unrolled explicit network (Bai et al., 2019; Winston & Kolter, 2020):

$$z_k = \sigma(W z_{k-1} + v) \tag{3.1}$$

with $\lim_{k\to\infty} z_k = z^\star$. This equivalence is informal and mainly serves to motivate our derivation of the IBP-MonDEQ. Given interval bounds $\bar{v}$ and $\underline{v}$ on $v$ satisfying $\underline{v} \leqslant v \leqslant \bar{v}$, where the inequalities hold elementwise, we can also follow IBP and (2.3) to iteratively obtain interval bounds on $z_k$:

$$\begin{bmatrix} \bar{z}_k \\ \underline{z}_k \end{bmatrix} = \sigma\left(\begin{bmatrix} (W)_+ & (W)_- \\ (W)_- & (W)_+ \end{bmatrix}\begin{bmatrix} \bar{z}_{k-1} \\ \underline{z}_{k-1} \end{bmatrix} + \begin{bmatrix} \bar{v} \\ \underline{v} \end{bmatrix}\right) \tag{3.2}$$

Just as we took $\lim_{k\to\infty} z_k$, we consider $\lim_{k\to\infty} \bar{z}_k, \underline{z}_k$, motivating another fixed-point problem:

$$\begin{bmatrix} \bar{z}^\star \\ \underline{z}^\star \end{bmatrix} = \sigma\left(\begin{bmatrix} (W)_+ & (W)_- \\ (W)_- & (W)_+ \end{bmatrix}\begin{bmatrix} \bar{z}^\star \\ \underline{z}^\star \end{bmatrix} + \begin{bmatrix} \bar{v} \\ \underline{v} \end{bmatrix}\right) \tag{3.3}$$

As shown in Figure 1, this IBP-inspired fixed point equation essentially maps the region $\{v' : \underline{v} \leqslant v' \leqslant \bar{v}\}$ to the region $\{z^{\star\prime} : \underline{z}^\star \leqslant z^{\star\prime} \leqslant \bar{z}^\star\}$, where inequalities are coordinate-wise. If they exist, the fixed points $\bar{z}^\star$ and $\underline{z}^\star$ will indeed lead to valid interval bounds (as defined in (2.2)) on $z^\star$. Proposition 3.2 states this observation formally.

### 3.1    FIXED POINTS OF THE IBP-INSPIRED EQUILIBRIUM EQUATION

Following the derivation above, the IBP fixed point equation (3.3) can be interpreted as the result of applying IBP to an infinitely deep, weight-tied neural network. However, IBP is notoriously unstable for deeper networks, so it is unclear whether a fixed point solution to (3.3) always exists. This section characterizes a class of $W$ for which (3.3) does have a fixed point (Theorem 3.1).

Figure 2 (left) demonstrates that the performance of IBP degrades for deep networks. We add linear layers to the architecture proposed by Gowal et al. (2018) for IBP, which is a wide but shallow network with 6 linear layers, and find that the final error certified by IBP increases with the depth of the model. This illustrates the challenge of getting IBP to work for deep models.

The challenges associated with depth also apply to the class of MonDEQs proposed by (Winston & Kolter, 2020). We will define $\widehat{W} \triangleq \begin{bmatrix} (W)_+ & (W)_- \\ (W)_- & (W)_+ \end{bmatrix} \in \mathbb{R}^{2h \times 2h}$ to be the linear transformation for IBP. Recall that Proposition 2.1 demonstrates existence and uniqueness of a fixed point solution to (2.4) when $I_{h \times h} - W$ is PD, but this does not guarantee the positive-definiteness of $I_{2h \times 2h} - \widehat{W}$, which would be sufficient for a fixed point solution to (3.3). Indeed, Figure 2 (right) shows that the fixed-point solver fails to solve (3.3) for the standard MonDEQ, as the interval bounds diverge. Intuitively, this failure occurs because the spectral radius of $\widehat{W}$ is always larger than that of $W$ (see Lemma 3.4), so repeated application of (3.2) diverges. Thus, further restrictions on $W$ are required.

The following theorem exactly characterizes the class of $W$ for which $I_{2h \times 2h} - \widehat{W}$ is PD. For these $W$, existence and uniqueness of fixed points to (2.4) and (3.3) is guaranteed to hold.

**Theorem 3.1.** *In the setting above, $I_{2h \times 2h} - \widehat{W} > 0$ if and only if $W$ is parameterized as follows:*

$$W = D^{-1/2} M D^{-1/2} \tag{3.4}$$

*where $D, M \in \mathbb{R}^{h \times h}$ and $D$ is a diagonal matrix satisfying $D_{i,i} > \frac{(|M|\mathbf{1})_i + (|M|^\top \mathbf{1})_i}{2}$, where $|M| \in \mathbb{R}^{h \times h}$ takes the entry-wise absolute value of $M$. Thus, for $W$ parameterized by (3.4), unique solutions $z^\star, \bar{z}^\star, \underline{z}^\star$ to the standard (2.4) and IBP (3.6) equilibrium equations exist $\forall\, v, \bar{v}, \underline{v} \in \mathbb{R}^h$.*

Theorem 3.1 is our main theoretical result, and proposes a simple class of $W$ for which the IBP equilibrium equation has a guaranteed unique fixed point. In this characterization, $M$ is an unconstrained matrix (and is also treated as an unconstrained parameter in our implementation, as described in Section 4). In Section 3.3, we theoretically analyze the expressivity of the induced class of DEQ layers. The proof of Theorem 3.1 is discussed in Section 3.2.

Next, it remains to check that $\bar{z}^\star$ and $\underline{z}^\star$ do provide valid interval bounds on $z^\star$, as the derivation in Section 3 was only heuristic. The following proposition makes this derivation rigorous.

**Proposition 3.2.** *In the setting above, suppose $\widehat{W}$ satisfies $I_{2h \times 2h} - \widehat{W} > \mathbf{0}$, so that unique fixed-point solutions $\bar{z}^\star, \underline{z}^\star$ to (3.3) always exist. Define the fixed-point solution $z^\star(v)$ as a function of $v$ as follows: $z^\star(v) = \sigma(W z^\star(v) + v)$. Then for any $\underline{v} \leqslant \bar{v}$ and all $i$, it must hold that*

$$(\underline{z}^\star)_i \leqslant \min_{\underline{v} \leqslant v \leqslant \bar{v}} (z^\star(v))_i \leqslant \max_{\underline{v} \leqslant v \leqslant \bar{v}} (z^\star(v))_i \leqslant (\bar{z}^\star)_i \tag{3.5}$$

We present the proof in Section A. Combining Theorem 3.1 and Proposition 3.2, we finally obtain the IBP-MonDEQ layer, which simultaneously solves two equilbrium equations to output both $z^\star$ and valid upper and lower interval bounds $\bar{z}^\star, \underline{z}^\star$ on $z^\star$:

$$z^\star = \sigma(W z^\star + v(x))$$
$$\begin{bmatrix} \bar{z}^\star \\ \underline{z}^\star \end{bmatrix} = \sigma\left( \begin{bmatrix} (W)_+ & (W)_- \\ (W)_- & (W)_+ \end{bmatrix} \begin{bmatrix} \bar{z}^\star \\ \underline{z}^\star \end{bmatrix} + \begin{bmatrix} \bar{v}(x, \epsilon) \\ \underline{v}(x, \epsilon) \end{bmatrix} \right) \tag{3.6}$$

To include an IBP-MonDEQ layer in a deeper model, we set $v(x)$ to be the output of the preceding layer, and choose $\bar{v}(x, \epsilon), \underline{v}(x, \epsilon)$ to be upper and lower interval bounds on $v(x)$. The interval bounds $\bar{z}^\star$ and $\underline{z}^\star$ can be propagated through the rest of the network following the standard IBP methods.

**LBEN parameterization.** We can relax the restrictions imposed by the parameterization (3.4) by leveraging the Lipschitz-bounded equilibrium network (LBEN) parameterization proposed by Revay et al. (2020). Revay et al. (2020) show that a guaranteed unique fixed point of (2.4) still exists if the PD condition on $I - W$ is relaxed to require $\Lambda - \Lambda W > \mathbf{0}$ for $\Lambda \in \mathbb{D}_+^{h \times h}$, the set of positive diagonal matrices. The following result exactly characterizes $W$ such that $\exists \Lambda \in \mathbb{D}_+^{h \times h} : \Lambda - \Lambda |W| > \mathbf{0}$, which is also sufficient for guaranteeing a unique fixed point solution to (3.3).

**Theorem 3.3.** *For weight matrix $W$, $\exists \Lambda \in \mathbb{D}_+^{h \times h}$ such that $\Lambda - \Lambda |W| > \mathbf{0}$ if and only if $W$ can be parameterized as follows:*

$$W = \Gamma D^{-1/2} M D^{-1/2} \Gamma^{-1} \tag{3.7}$$

*for some $\Gamma, D \in \mathbb{D}_+^{h \times h}$ with $D$ satisfying $D_{i,i} > \frac{(|M|\mathbf{1}+|M|^\top\mathbf{1})}{2}$.*

*For $W$ parameterized by (3.7), we have the following immediate consequences: 1) unique solutions $z^\star, \bar{z}^\star, \underline{z}^\star$ to the equilbrium equations (3.6) exist for all $v, \bar{v}, \underline{v} \in \mathbb{R}^h$, and 2) $\bar{z}^\star, \underline{z}^\star$ are upper and lower interval bounds on $z^\star$ in the sense of Proposition 3.2.*

The proof is provided in Section A. Note that Theorem 3.3 improves the expressivity of the IBP-MonDEQ layer because the class of $W$ proposed in Theorem 3.3 contains that of Theorem 3.1. In Section 3.3, we show that, perhaps surprisingly, this more flexible parameterization allows IBP-MonDEQ layers to express all explicit networks. Section 4 demonstrates that this parameterization can also lead to empirical improvements.

## 3.2 PROOF OF THEOREM 3.1

We sketch the proof of Theorem 3.1. To prove $I_{2h \times 2h} - \widehat{W} > 0$, we require a simple way to reason about the spectrum of $\widehat{W}$. The following lemma formalizes a connection between eigenvalues of $\frac{\widehat{W}+\widehat{W}^\top}{2}$ and $\frac{|W|+|W|^\top}{2}$.

**Lemma 3.4.** *All eigenvalues of $\frac{\widehat{W}+\widehat{W}^\top}{2}$ are either eigenvalues of $\frac{W+W^\top}{2}$ or $\frac{|W|+|W|^\top}{2}$. As a direct consequence, the following equivalence holds:*

$$I_{2h \times 2h} - \widehat{W} > \mathbf{0} \iff I_{h \times h} - |W| > \mathbf{0} \tag{3.8}$$

Note that the condition that $I_{h \times h} - |W|$ is PD is a stricter requirement than requiring positive-definiteness of $I_{h \times h} - W$, as it always holds that $\lambda_{\max}(\frac{|W|+|W|^\top}{2}) \geqslant \lambda_{\max}(\frac{W+W^\top}{2})$, where $\lambda_{\max}$ denotes the largest eigenvalue. Thus, in order for both (2.4) and (3.3) to have unique fixed points, it suffices to have $I_{h \times h} - |W| > 0$. The proof of Lemma 3.4 is completed in Section A and relies on the following basic fact about structured block matrices.

**Claim 3.5.** *Let $M \in \mathbb{R}^{2h \times 2h}$ be any matrix of the form $M = \begin{bmatrix} A & B \\ B & A \end{bmatrix}$, where $A, B \in \mathbb{R}^{h \times h}$. Then all eigenvalues of $M$ are either eigenvalues of $A + B$ or $A - B$.*

Following Lemma 3.4, we now complete the proof of Theorem 3.1 by analyzing $I_{h \times h} - |W|$.

*Proof of Theorem 3.1.* By Lemma 3.4, it is equivalent to characterize when $I_{h \times h} - |W| > \mathbf{0}$. We first show that (3.4) implies positive-definiteness. We write

$$I - \frac{|W| + |W|^\top}{2} = D^{-1/2}\left(D - \frac{|M| + |M|^\top}{2}\right)D^{-1/2} \tag{3.9}$$

By the lower bound on entries of $D$, the matrix $D - \frac{|M|+|M|^\top}{2}$ is symmetric and strictly diagonally dominant (SDD), and thus PD. Thus, we obtain $I - \frac{|W|+|W|^\top}{2} > 0$, so $I - |W| > 0$.

For the reverse direction, we appeal to the theory of M-matrices (Plemmons, 1977). We note that $I - \frac{|W|+|W|^\top}{2}$ is a nonsingular M-matrix, i.e., a matrix with negative off-diagonal entries whose eigenvalues have positive real parts. By an equivalent characterization of M-matrices, there exists a diagonal matrix $E \in \mathbb{D}_+$ such that $E(I - \frac{|W|+|W|^\top}{2})E$ is SDD. Rearranging gives $|W| = I - E^{-1}AE^{-1} + S$, where $A$ is symmetric and SDD, and $S$ is skew-symmetric ($S^\top = -S$).

Now we choose $M$ satisfying $|M| = E^2 - A + ESE$ and $D = E^2$, so that $W$ has the parameterization $W = D^{-1/2}MD^{-1/2}$. To check that $D_{i,i} > \frac{(|M|\mathbf{1})_i+(|M|^\top\mathbf{1})_i}{2}$, we note that $\frac{(|M|\mathbf{1})_i+(|M|^\top\mathbf{1})_i}{2} = ((E^2 - A)\mathbf{1})_i = E_{i,i}^2 - A_{i,i} - \sum_{j \neq i} A_{i,j} < E_{i,i}^2$, where we use the fact that $A_{i,i} > \sum_{j \neq i} |A_{i,j}|$ because $A$ is SDD. Plugging in $D_{i,i} = E_{i,i}^2$ gives the desired inequality. $\square$

## 3.3 EXPRESSIVITY OF THE IBP-MONDEQ

Guaranteeing that (3.3) has a fixed point requires placing further restrictions on $W$ on top of those already proposed for MonDEQs (Winston & Kolter, 2020). This section examines the expressivity

of the resulting IBP-MonDEQ parameterization. To sanity check that our parameterization is not too restrictive, we first contrast against normalization techniques for controlling Lipschitz constants in neural networks. Next, we demonstrate that with the relaxed parameterization of Theorem 3.3, the IBP-MonDEQ is able to express traditional explicit feedforward networks.

**Operator norm of $W$.** We first investigate the operator norm of $W$. A number of papers constrain the weight matrices so that their operator norms are bounded by 1, which prevents blowup of the naive upper bound $\prod_i \|W_i\|_{\text{op}}$ on the Lipschitz constant of the whole network (Cisse et al., 2017; Miyato et al., 2018; Farnia et al., 2018; Trockman & Kolter, 2021). This approach has applications in robustness, but there are also concerns that constraining the operator norms to 1 severely harms expressivity (Huster et al., 2018; Anil et al., 2019). We verify that the parameterization of Theorem 3.1 *does not* imply the stringent constraint $\|W\|_{\text{op}} \leqslant 1$.

**Example 3.6.** *The following choice of $W$ satisfies the parameterization* (3.4) *with* $\|W\|_{\text{op}} = 1.9$*:* $W = \begin{bmatrix} 0 & 1.9 \\ 0.05 & 0 \end{bmatrix}$. *We set $M = W$ and $D = I_{2\times 2}$, so $W = D^{-1/2}MD^{-1/2}$ as in* (3.4).

Example 3.6 may be surprising given superficial similarities between (3.4) and the following parameterization, which does imply $\|W\|_{\text{op}} \leqslant 1$: $W = D^{-1}M$, where $D$ satisfies $D_{i,i} \geqslant (|M|\mathbf{1})_i$. The parameterization in Theorem 3.1 avoids this constraint because the requirement is instead $D_{i,i} \geqslant \frac{(|M|\mathbf{1} + |M|^\top \mathbf{1})_i}{2}$, so mass in $M$ can be asymmetrically distributed between the upper and lower triangular halves, as in Example 3.6.

**Expressing explicit feedforward networks.** The following result states that using the parameterization in Theorem 3.3, a single IBP-MonDEQ layer can express all explicit feedforward networks.

**Proposition 3.7.** *Consider explicit neural net embedding functions of the following form:*

$$z_L(x) = \sigma(A_L \sigma(A_{L-1} \cdots \sigma(A_1 x) \cdots)) \tag{3.10}$$

*where $z_L : \mathbb{R}^d \to \mathbb{R}^h$ and $A_i \in \mathbb{R}^{h \times h} \, \forall 1 \leqslant i \leqslant L$. There is a DEQ layer $z^\star : \mathbb{R}^d \to \mathbb{R}^{Lh}$ outputting $z^\star(x) = \sigma(Wz^\star(x) + Bx)$, where $W, B \in \mathbb{R}^{Lh \times Lh}$, such that $W$ obeys the parameterization of Theorem 3.3, and the last $h$ coordinates of $z^\star$ satisfy*

$$(z^\star(x))_{(L-1)h:Lh} = z_L(x) \, \forall x \in \mathbb{R}^d$$

Proposition 3.7 states that the IBP-MonDEQ parameterized in Theorem 3.3 is at least as powerful as standard explicit networks, and is somewhat surprising given that the parameterizations (3.4) and (3.7) may appear restrictive. The proof, provided in Section A, leverages the fact that $\Gamma$ provides flexibility in scaling the weights, essentially allowing us to cancel out the normalization matrix $D$.

# 4 EXPERIMENTS

In this section, we empirically compare models with IBP-MonDEQ layers against fully explicit models. Our results show that IBP-MonDEQ models can achieve comparable performance on common certified robustness benchmarks relative to IBP with standard explicit models.

**Explicit models.** Table 1 illustrates the explicit architectures: a smaller 3LAYER model with 2 convolutional layers and 128 output channels, and a larger 7LAYER model. The 7LAYER model is a wide and shallow network with 17 million parameters proposed by Gowal et al. (2018) for IBP.

**IBP-MonDEQ models.** As shown in Table 1, we consider replacing a convolutional layer in a fully-explicit model with an IBP-MonDEQ layer where $W$ computes an equivalently-sized convolutional transformation. This leads to DEQ-3 and DEQ-7 models with equivalent parameter counts as their fully-explicit counterparts. We use +LBEN to indicate that the IBP-MonDEQ layer uses the LBEN parameterization in Theorem 3.3. To solve for the fixed points, we use the forward-backward algorithm with Anderson acceleration (Walker & Ni, 2011; imp, 2020).

To guarantee that (3.6) has a unique fixed point, we enforce either (3.4) or (3.7) (depending on whether we use the LBEN parameterization) for the IBP-MonDEQ weight $W$. To enforce (3.4), we set $W = D^{-1/2}MD^{-1/2}$ where $M$ is the only trainable parameter and $D = \text{diag}\left(\frac{(|M| + |M|^\top)\mathbf{1}}{2(1-m)}\right)$ is a fixed function of $M$, where $\text{diag}(\cdot)$ maps a vector to a diagonal matrix. Here $m \in (0,1)$ is a

Table 1: **Illustration of architectures.** CONV-$h$-$k$x$k$-$s$ denotes a convolution with $h$ output channels, $k$x$k$ kernel size, and $s$ stride. We consider DEQ models where a CONV layer is replaced by a IBPMON-$h$-$k$x$k$-$s$ layer with an equivalently-sized convolutional operation.

| 3LAYER | DEQ-3 | 7LAYER | DEQ-7 |
|---|---|---|---|
| | | CONV-64-3x3-1 | CONV-64-3x3-1 |
| | | CONV-64-3x3-1 | **IBPMON-64-3x3-1** |
| CONV-128-3x3-1 | CONV-128-3x3-1 | CONV-128-3x3-2 | CONV-128-3x3-2 |
| CONV-128-3x3-1 | **IBPMON-128-3x3-1** | CONV-128-3x3-1 | CONV-128-3x3-1 |
| FC-10 | FC-10 | CONV-128-3x3-1 | CONV-128-3x3-1 |
| | | FC-512 | FC-512 |
| | | FC-10 | FC-10 |

Table 2: **Certified robustness results.** We display the average certified error and standard (clean) error for explicit and IBP-MonDEQ architectures (lower is better). We compute the mean errors over 3 independent training runs, with standard deviations in parentheses. Our results show that the IBP-MonDEQ models can achieve smaller certifiably robust error than their fully explicit counterparts.

| | | MNIST | | | | CIFAR10 | |
|---|---|---|---|---|---|---|---|
| | | $\epsilon = 0.1$ | $\epsilon = 0.2$ | $\epsilon = 0.3$ | $\epsilon = 0.4$ | $\epsilon = \frac{2}{255}$ | $\epsilon = \frac{8}{255}$ |
| Certified error | 3LAYER | 4.19 (0.27) | 9.16 (0.17) | 19.14 (0.39) | 28.98 (0.41) | **53.01** (0.45) | 72.38 (0.35) |
| | DEQ-3 | 3.18 (0.09) | **6.56** (0.11) | 12.72 (0.49) | 19.80 (0.66) | 53.9 (0.37) | **71.97** (0.31) |
| | (+LBEN) | **3.15** (0.07) | 6.64 (0.13) | **11.96** (0.38) | **19.11** (0.32) | 54.18 (0.12) | 72.33 (0.59) |
| Standard (Clean) error | 3LAYER | 2.10 (0.06) | 3.33 (0.21) | 6.54 (0.09) | 6.49 (0.12) | **39.22** (0.16) | **58.66** (0.15) |
| | DEQ-3 | 1.62 (0.03) | 2.62 (0.11) | 4.71 (0.27) | **4.44** (0.20) | 40.25 (0.39) | 59.35 (0.99) |
| | (+LBEN) | **1.60** (0.04) | **2.49** (0.12) | **4.54** (0.12) | 4.58 (0.31) | 40.77 (0.29) | 59.41 (0.29) |
| Certified error | 7LAYER | 2.61 (0.11) | 4.21 (0.26) | 7.39 (0.09) | 12.41 (0.27) | 48.46 (0.31) | 66.95 (0.35) |
| | DEQ-7 | 2.58 (0.06) | 4.24 (0.07) | **6.80** (0.04) | 12.32 (0.37) | 48.29 (0.28) | **66.87** (0.34) |
| | (+LBEN) | **2.51** (0.03) | **4.15** (0.04) | 6.92 (0.14) | **12.25** (0.03) | **48.23** (0.34) | 67.20 (0.37) |
| Standard (clean) error | 7LAYER | 0.89 (0.02) | 1.27 (0.04) | 2.25 (0.19) | 2.29 (0.15) | **32.82** (0.33) | **52.84** (1.30) |
| | DEQ-7 | **0.79** (0.06) | 1.16 (0.03) | 2.13 (0.05) | 2.13 (0.07) | 33.04 (0.49) | 53.12 (0.88) |
| | (+LBEN) | 0.80 (0.04) | **1.13** (0.02) | **2.04** (0.05) | **2.10** (0.07) | 33.08 (0.48) | 53.42 (0.58) |

tunable hyperparameter governing a tradeoff between expressivity and convergence rate: a smaller $m$ leads to more expressivity but also slower convergence of the equilbrium solvers. For the LBEN parameterization (3.7), we parameterize the additional diagonal scaling matrix $\Gamma \in \mathbb{D}_+^{h \times h}$ by $\Gamma = \text{diag}\left(\frac{\gamma-1}{\gamma}\text{sigmoid}(\beta) + \frac{1}{\gamma}\right)$, where $\gamma > 1$ is a scalar hyperparameter, and $\beta \in \mathbb{R}^h$ is a trainable parameter $\beta \in \mathbb{R}^h$. This parameterization enforces that the condition number of $\Gamma$ is bounded by $\gamma$.

## 4.1 CERTIFIED ROBUSTNESS RESULTS

Tables 2 shows certified and standard classification errors of 3 and 7 layer models trained with IBP on the MNIST and CIFAR10 datasets for various values of $\epsilon$. For most settings, IBP-MonDEQ models are able to achieve lower mean certified errors than their fully explicit counterparts.

The improvement is particularly striking in the case of 3 layer models on MNIST, where IBP-MonDEQ models perform much better, e.g. 9.87% better on MNIST for $\epsilon = 0.4$. We also find that using the more relaxed LBEN parameterization (Theorem 3.3) generally leads to better performance on MNIST, but the reverse trend may hold for CIFAR10. We hypothesize that the LBEN parameterization may lead to optimization difficulties in the IBP-MonDEQ layer. Investigating and improving the optimization of IBP-MonDEQ layers is an interesting direction for future work.

**Implementation details.** We train using IBP with the Adam optimizer (Kingma & Ba, 2014) with a learning rate of 5e-4, and report errors at the last epoch of training averaged over 3 runs. We use a smooth warmup schedule for $\epsilon$ (Xu et al., 2020; Shi et al., 2021), interpolating between $\epsilon = 0$ and $\epsilon = \epsilon_{\text{train}}$, a target training $\epsilon$ lower bounded by $\epsilon_{\text{test}}$.

Following Shi et al. (2021), we use batch normalization (BN) after every linear layer besides the last FC-10 layer. There is no BN within an IBP-MonDEQ layer, but BN is applied to outputs of the IBP-MonDEQ layers. For 3 layer models, we insert a 4×4 pooling layer before the FC-10 layer. Additional details are in Section B.

## 5 ADDITIONAL RELATED WORKS

**Implicit layers.** Various fundamental properties of DEQs have been investigated in the literature, most notably convergence to and existence of a fixed point (Winston & Kolter, 2020; Revay et al., 2020; Bai et al., 2021). Bai et al. (2021) propose empirical Jacobian regularization techniques for speeding up equilibrium solving in DEQs. Another studied property of DEQs is their Lipschitzness: Revay et al. (2020) and Pabbaraju et al. (2020) derive bounds on the Lipschitz constants of MonDEQs which are applicable in certified robustness settings. However, because the Lipschitz constants are tailored to the $\ell_2$ norm, these works only consider $\ell_2$ bounded adversaries. Chen et al. (2021) propose polynomial optimization programs for certifying robustness of MonDEQs, but their methods only succeed for $\ell_2$ perturbations and do not work empirically against $\ell_\infty$ adversaries. Kawaguchi (2021) theoretically analyze optimization of equilibrium models.

Other prominent classes of implicit models include neural ODEs (Chen et al., 2018; Dupont et al., 2019; Liu et al., 2019; Finlay et al., 2020), and convex optimization layers (Amos et al., 2017; Amos & Kolter, 2017; Agrawal et al., 2019). Neural ODE outputs are implicitly defined as the solution to some differential equation. Adversarial robustness is also an interesting topic of study for neural ODEs, which may be more robust than explicit models (Yan et al., 2019), though the cause of this might be gradient obfuscation (Huang et al., 2020).

**Certified robustness.** A common technique in certified robustness is to express a convex program which lower bounds the worst-case perturbed output of the network, either via semidefinite relaxations (Raghunathan et al., 2018a;b), or linear relaxations (Wong & Kolter, 2018; Wong et al., 2018; Dvijotham et al.; Zhang et al., 2018; Wang et al., 2018). One advantage of IBP compared to these approaches is its simplicity and efficiency, making it viable for training larger models (Gowal et al., 2018). Zhang et al. (2019) propose CROWN-IBP, which combines IBP with linear relaxation methods during warmup training. Xu et al. (2020) extend this approach to general computation graphs. Shi et al. (2021) propose methods for speeding up optimization of IBP-trained networks.

Another well-studied certified robustness approach is randomized smoothing (Li et al., 2018; Cohen et al., 2019; Lecuyer et al., 2019; Salman et al., 2019a; Yang et al., 2020), which can in theory produce probabilistic certificates of robustness for generic models (including, in theory, DEQs). However, these certificates are random, not deterministic, and have some error probability. In addition, randomized smoothing has theoretical limitations for certifying $\ell_\infty$ robustness for high-dimensional data (Blum et al., 2020; Kumar et al., 2020; Yang et al., 2020).

## 6 CONCLUSION

This work aims to certify robustness of DEQ models. We propose the IBP-MonDEQ layer, a DEQ layer which solves two equilibrium equations: one for the unperturbed fixed-point output, and one for upper and lower interval bounds on adversarially perturbed values of the fixed point. Our mathematical analysis reveals how to parameterize the IBP-MonDEQ so that existence and uniqueness of a fixed point is guaranteed. One interesting interpretation of these results is that they characterize a class of infinitely deep, weight-tied neural nets for which IBP is stable. An interesting direction for future work is to explore whether other methods besides IBP for certifying robustness of explicit networks can also be adapted to work for DEQs.

## ACKNOWLEDGEMENTS

We thank Huan Zhang for helpful discussions and Alex Wang for helpful comments on a draft of this work. CW was supported by a NSF Graduate Research Fellowship. Portions of this work were supported by funds from Toyota Research Institute and the Bosch Center for AI.

## ETHICS AND REPRODUCIBILITY STATEMENTS

We do not expect ethical concerns to arise from this research because DEQ methods are still exploratory in nature, and so the main impacts of this work are still mainly theoretical and algorithmic.

All proofs omitted from the main body of the paper are in Section A. Additional implementation details are provided in Section B, and code is available at the following link: `https://github.com/cwein3/ibp-mondeq-code`.

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

## A  MISSING PROOFS IN SECTION 3

We first fill in the proofs of Claim 3.5 and Lemma 3.4.

*Proof of Claim 3.5.* Let $u = \begin{bmatrix} s \\ t \end{bmatrix}$ be any eigenvector of $M$, with eigenvalue $\lambda$. There are two cases:

**Case 1:** $s \neq -t$. As $\lambda \begin{bmatrix} s \\ t \end{bmatrix} = Mu = \begin{bmatrix} As + Bt \\ Bs + At \end{bmatrix}$, it follows that $\lambda s = As + Bt$ and $\lambda t = Bs + At$. Thus, it follows that $(A + B)(s + t) = As + Bt + Bs + At = \lambda(s + t)$. As the condition of this case assumed $(s + t) \neq \mathbf{0}$, $\lambda$ is an eigenvalue of $A + B$.

**Case 2:** $s = -t$. Then $\lambda \begin{bmatrix} s \\ -s \end{bmatrix} = Mu = \begin{bmatrix} (A - B)s \\ -(A - B)s \end{bmatrix}$. It follows that $\lambda s = (A - B)s$, so $\lambda$ is an eigenvalue of $A - B$. $\qquad \square$

Using Claim 3.5, we can complete the proof of Lemma 3.4.

*Proof of Lemma 3.4.* Define $A \triangleq \frac{(W)_+ + (W)_+^\top}{2}$ and $B \triangleq \frac{(W)_- + (W)_-^\top}{2}$, so $\widehat{W} = \begin{bmatrix} A & B \\ B & A \end{bmatrix}$. We note that $A + B = \frac{((W)_+ + (W)_-) + ((W)_+ + (W)_-)^\top}{2} = \frac{W + W^\top}{2}$ and likewise $A - B = \frac{|W| + |W|^\top}{2}$. We now directly apply Claim 3.5 to conclude that eigenvalues of $\widehat{W}$ are either eigenvalues of $A + B$ or $A - B$, as desired.

To see the left-to-right implication of (3.8), we use the fact that the set of eigenvalues of $\widehat{W} + \widehat{W}^\top$ contains those of $|W| + |W|^\top$ as a subset. For the right-to-left implication, we note that $I - |W| > 0$ implies that $\lambda_{\max}((|W| + |W|^\top)/2) < 1$. Next, we can observe that $\lambda_{\max}(|W| + |W|^\top) = \lambda_{\max}(\widehat{W} + \widehat{W}^\top)$ because eigenvalues of $\widehat{W} + \widehat{W}^\top$ are either eigenvalues of $W + W^\top$ or eigenvalues of $|W| + |W|^\top$. Now we can use the fact that $\lambda_{\max}(W + W^\top) \leqslant \lambda_{\max}(|W| + |W|^\top)$, which would give us $\lambda_{\max}((\widehat{W} + \widehat{W}^\top)/2) < 1$, or $I - \widehat{W} > 0$.

Finally, to see that $\lambda_{\max}(W + W^\top) \leqslant \lambda_{\max}(|W| + |W|^\top)$, let $v$ be the eigenvector corresponding to the maximum eigenvalue of $W + W^\top$. The key observation is that $v^\top(W + W^\top)v \leqslant |v|^\top(|W| + |W|^\top)|v|$ because terms in the expansion of the first quantity will be replaced by positive versions with the same magnitude in the r.h.s. $\qquad \square$

Next, we complete the proof of Proposition 3.2.

*Proof of Proposition 3.2.* By results of Winston & Kolter (2020), there exists $0 < \alpha \leqslant 1$ such that the damped iteration

$$
\begin{aligned}
\begin{bmatrix} \bar{u}_k \\ \underline{u}_k \end{bmatrix} &= (1 - \alpha) \begin{bmatrix} \bar{z}_{k-1} \\ \underline{z}_{k-1} \end{bmatrix} + \alpha \left( \widehat{W} \begin{bmatrix} \bar{z}_{k-1} \\ \underline{z}_{k-1} \end{bmatrix} + \begin{bmatrix} \bar{v} \\ \underline{v} \end{bmatrix} \right) \\
\begin{bmatrix} \bar{z}_k \\ \underline{z}_k \end{bmatrix} &= \sigma \left( \begin{bmatrix} \bar{u}_k \\ \underline{u}_k \end{bmatrix} \right)
\end{aligned}
\tag{A.1}
$$

is guaranteed to converge for any initial $\bar{z}_0, \underline{z}_0$: $\lim_{k \to \infty} \bar{z}_k, \underline{z}_k = \bar{z}^\star, \underline{z}^\star$. For any $v$ satisfying $\underline{v} \leqslant v \leqslant \bar{v}$, we consider iterating (A.1) with $\bar{z}_0 = \underline{z}_0 = z^\star(v)$.

We claim that the following invariant holds: $\bar{z}_k \geqslant z^\star(v) \geqslant \underline{z}_k$ $\forall k \geqslant 0$. To see this, we also consider the damped iteration for $z^\star(v)$, noting that the following equality holds by (Winston & Kolter, 2020):

$$
\begin{aligned}
u^\star &= (1 - \alpha)z^\star(v) + \alpha(Wz^\star(v) + v) \\
z^\star(v) &= \sigma(u^\star)
\end{aligned}
\tag{A.2}
$$

Now we apply induction on $k$. The base case $k = 0$ given by our choice of $\bar{z}_0, \underline{z}_0$. We compare corresponding terms in (A.1) and (A.2) for computing $\bar{u}, \underline{u}$, and $u^\star$. We note that $\bar{z}_{k-1} \geqslant z^\star(v)$ by the inductive hypothesis, $\left( \widehat{W} \begin{bmatrix} \bar{z}_{k-1} \\ \underline{z}_{k-1} \end{bmatrix} \right)_{0:h} \geqslant z^\star(v)$ by the inductive hypothesis and properties of

the IBP iteration, and $\bar{v} \geqslant v$ by assumption. Thus, it follows that $\bar{u}_k \geqslant u^\star$, and likewise, $\underline{u}_k \leqslant u^\star$. Since $\sigma$ is nondecreasing, we obtain $\bar{z}_k \geqslant z^\star(v) \geqslant \underline{z}_k$, completing the induction. Taking limits as $k \to \infty$, we obtain $\bar{z}^\star \geqslant z^\star(v) \geqslant \underline{z}^\star$, as desired. $\qquad\square$

Next, we complete the proof of Theorem 3.3. The following lemma will be useful.

**Lemma A.1.** *In the setting of Theorem 3.3, suppose that $W$ satisfies $\Lambda - \Lambda W > \mathbf{0}$ for some $\Lambda \in \mathbb{D}_+^{h \times h}$. Then there exists $0 < \alpha \leqslant 1$ such that the following iteration converges:*

$$z_k = \sigma((I - \alpha\Lambda)z_{k-1} + \alpha\Lambda(Wz_{k-1} + v)) \tag{A.3}$$

*with $\lim_{k\to\infty} z_k = z^\star$, where $z^\star$ is the solution to the fixed point equation $z^\star = \sigma(Wz^\star + v)$.*

*Proof of Lemma A.1.* We first observe that the linear operator $I - \alpha(\Lambda - \Lambda W)$ is contractive for sufficiently small $\alpha$, as $\Lambda - \Lambda W > \mathbf{0}$. Thus, since ReLU is 1-Lipschitz, the iteration in (A.3) contracts and therefore converges to a fixed point. To check that this gives the same fixed point as (3.3), we will check that the following holds:

$$z^\star = \sigma((I - \alpha\Lambda)z^\star + \alpha\Lambda(Wz^\star + v)) \tag{A.4}$$

Note that for coordinates $i$ where $(Wz^\star + v)_i > 0$, the definition of $z^\star$ implies $z_i^\star = (Wz^\star + v)_i$, so (A.4) is verified for such coordinates. Otherwise, for coordinates $i$ where $(Wz^\star + v)_i \leqslant 0$, since $\sigma$ is the ReLU activation, $z_i^\star = 0$, so we can again verify (A.4) for these coordinates. $\qquad\square$

We now complete the proof of Theorem 3.3.

*Proof of Theorem 3.3.* First, we consider any $W$ of the form (3.7). Taking $\Lambda = \Gamma^{-2}$, we have $\Lambda - \Lambda W = \Gamma^{-1}(I - D^{-1/2}MD^{-1/2})\Gamma^{-1}$. Now by invoking Theorem 3.1, we conclude that the middle term in this product is PD, so $\Lambda - \Lambda W > \mathbf{0}$.

To prove the reverse implication, we note that $\Lambda - \Lambda|W| = \Lambda^{1/2}(I - \Lambda^{1/2}|W|\Lambda^{-1/2})\Lambda^{1/2}$, so $\Lambda - \Lambda|W| > \mathbf{0} \iff I - \Lambda^{1/2}|W|\Lambda^{-1/2} > \mathbf{0}$. By invoking Theorem 3.1, it follows that we have the parameterization $\Lambda^{1/2}W\Lambda^{-1/2} = D^{1/2}MD^{1/2}$, so setting $\Gamma = \Lambda^{-1/2}$ and rearranging gives the desired result.

Next, we check that (2.4) and (3.3) have guaranteed unique fixed points whenever $W$ is parameterized as in (3.7). Using the observation derived above, we can check that

$$\Lambda - \Lambda|W| > \mathbf{0} \tag{A.5}$$
$$\implies I - \Lambda^{1/2}|W|\Lambda^{-1/2} > \mathbf{0} \tag{A.6}$$
$$\implies I - \Lambda^{1/2}W\Lambda^{-1/2} > \mathbf{0} \tag{A.7}$$
$$\implies \Lambda - \Lambda W > \mathbf{0} \tag{A.8}$$

Thus, we can apply Theorem 1 of (Revay et al., 2020) to conclude the existence of a unique fixed point for (2.4). For (3.3), we apply Lemma 3.4 to conclude that $\widehat{\Lambda} - \widehat{\Lambda}\widehat{W} > \mathbf{0}$, where $\widehat{\Lambda} \triangleq \begin{bmatrix} \Lambda & \mathbf{0} \\ \mathbf{0} & \Lambda \end{bmatrix} \in \mathbb{R}^{2h \times 2h}$. Thus, Theorem 1 of (Revay et al., 2020) can also be used to conclude that (3.3) has a unique fixed point.

Finally, it remains to check that $\bar{z}^\star$ and $\underline{z}^\star$ are indeed valid interval bounds on $z^\star$. The proof of Proposition 3.2 no longer directly applies because the damped iteration (A.1) might not converge, as we no longer have the guarantee $I_{2h \times 2h} - \widehat{W} > 0$. Instead, we invoke Lemma A.1, which states the existence of $0 < \alpha \leqslant 1$ such that the following iteration converges:

$$\begin{bmatrix} \bar{z}_k \\ \underline{z}_k \end{bmatrix} = \sigma\left( (I_{2h \times 2h} - \alpha\widehat{\Lambda}) \begin{bmatrix} \bar{z}_{k-1} \\ \underline{z}_{k-1} \end{bmatrix} + \alpha\widehat{\Lambda}\left( \widehat{W} \begin{bmatrix} \bar{z}_{k-1} \\ \underline{z}_{k-1} \end{bmatrix} + \begin{bmatrix} \bar{v} \\ \underline{v} \end{bmatrix} \right) \right) \tag{A.9}$$

with $\lim_{k\to\infty} \bar{z}_k = \bar{z}^\star$, $\lim_{k\to\infty} \underline{z}_k = \underline{z}^\star$. Using this modified iteration but keeping other steps the same, we can follow Proposition 3.2 to complete the proof. $\qquad\square$

Next, we provide the proof of Proposition 3.7.

*Proof of Proposition 3.7.* We set $B = \begin{bmatrix} A_1 & \mathbf{0} & \cdots & \mathbf{0} \\ \mathbf{0} & \mathbf{0} & \cdots & \mathbf{0} \\ \vdots & \vdots & \vdots & \vdots \\ \mathbf{0} & \mathbf{0} & \cdots & \mathbf{0} \end{bmatrix}$. According to (3.7), the weight matrix $W$ should take the following form: $W = \Gamma D^{-1/2} M D^{-1/2} \Gamma^{-1}$. We choose $\Gamma$ to take the following form:

$$\Gamma = \begin{bmatrix} \gamma_1 I_{h \times h} & \mathbf{0} & \cdots & \mathbf{0} \\ \mathbf{0} & \gamma_2 I_{h \times h} & \cdots & \mathbf{0} \\ \vdots & \vdots & \vdots & \vdots \\ \mathbf{0} & \cdots & \mathbf{0} & \gamma_L I_{h \times h} \end{bmatrix} \tag{A.10}$$

$M$ will take the following lower triangular block form:

$$M = \begin{bmatrix} \mathbf{0} & \cdots & \cdots & \cdots & \mathbf{0} \\ A_2 & \mathbf{0} & \cdots & \cdots & \mathbf{0} \\ \mathbf{0} & A_3 & \mathbf{0} & \cdots & \mathbf{0} \\ \vdots & \vdots & \vdots & \vdots & \vdots \\ \mathbf{0} & \cdots & \mathbf{0} & A_L & \mathbf{0} \end{bmatrix} \tag{A.11}$$

where if we split $M$ into a grid of $h \times h$ blocks, the block in the $i$-th row and $i-1$-th column has value $A_i$, and all other blocks are $\mathbf{0}$. Finally, $D$ will take the form:

$$D = \begin{bmatrix} \beta_1 I_{h \times h} & \mathbf{0} & \cdots & \mathbf{0} \\ \mathbf{0} & \beta_2 I_{h \times h} & \cdots & \mathbf{0} \\ \vdots & \vdots & \vdots & \vdots \\ \mathbf{0} & \cdots & \mathbf{0} & \beta_L I_{h \times h} \end{bmatrix} \tag{A.12}$$

In order to satisfy the constraints of (3.7), we require $\beta_1 > \max_{j \in [h]} \frac{(|A|_2^\top \mathbf{1})_j}{2}$, $\beta_L > \max_{j \in [h]} \frac{(|A|_L \mathbf{1})_j}{2}$, and $\beta_i > \max_{j \in [h]} \frac{((|A|_i + |A|_{i+1}^\top)\mathbf{1})_j}{2}$ for $2 \leqslant i \leqslant L - 1$. We now observe that $W$ is of the form

$$W = \begin{bmatrix} \mathbf{0} & \cdots & \cdots & \cdots & \mathbf{0} \\ \gamma_2 \beta_2^{-1/2} A_2 \beta_1^{-1/2} \gamma_1^{-1} & \mathbf{0} & \cdots & \cdots & \mathbf{0} \\ \mathbf{0} & \gamma_3 \beta_3^{-1/2} A_3 \beta_2^{-1/2} \gamma_2^{-1} & \mathbf{0} & \cdots & \mathbf{0} \\ \vdots & \vdots & \vdots & \vdots & \vdots \\ \mathbf{0} & \cdots & \mathbf{0} & \gamma_L \beta_L^{-1/2} A_L \beta_{L-1}^{-1/2} \gamma_{L-1}^{-1} & \mathbf{0} \end{bmatrix} \tag{A.13}$$

We observe now that we have $L$ degrees of freedom in choosing $\gamma$, so we can choose $\gamma$ such that $\gamma_i \beta_i^{-1/2} \beta_{i-1}^{-1/2} \gamma_{i-1}^{-1} = 1 \ \forall 2 \leqslant i \leqslant L$, as this only imposes $L-1$ constraints. For example, we can set $\gamma_1 = 1$, and set $\gamma_2, \ldots, \gamma_L$ in sequence so that the desired equalities hold. This gives $W = M$. Finally, it is straightforward to see that the equilibrium for this constructed $W$ and $B$ concatenates all of the hidden layers of the network, so the desired equivalence between the DEQ and explicit network holds. $\square$

## B  ADDITIONAL IMPLEMENTATION DETAILS

**Implementation details for Section 4.** For all models, we use a batch size of 128 and anneal the learning rate, which is initially 5e-4, by a factor of 0.2 at certain steps. For models trained on MNIST and CIFAR10 with $\epsilon = 2/255$, annealing occurs at the 120th and 280th epochs, and for CIFAR10 with $\epsilon = 8/255$, annealing occurs at the 240th and 280th epochs. We use gradient clipping with a max $\ell_2$ norm of 10. To preprocess the data, all inputs are normalized to the range $[0, 1]$ before we subtract the mean and divide by the standard deviation of the training images across each input channel. Note that the $\epsilon$ adversarial perturbations are applied to the $[0, 1]$-normalized image before subtracting the mean and dividing by the standard deviation. For MNIST, we do not use data

Table 3: $\epsilon_{\text{train}}$ v.s. $\epsilon_{\text{test}}$ for MNIST.

| $\epsilon_{\text{test}}$ | 0.1 | 0.2 | 0.3 | 0.4 |
|---|---|---|---|---|
| $\epsilon_{\text{train}}$ | 0.2 | 0.3 | 0.4 | 0.4 |

augmentation, and for CIFAR10 we use random crops and horizontal flips. We note that during both training and testing, when computing IBP bounds we elide the last FC-10 transformation with the output margins of the model (which can be expressed as a linear function of the logits), as Gowal et al. (2018) show that this leads to performance benefits.

For CIFAR10, we choose $\epsilon_{\text{train}} = \epsilon_{\text{test}}$, whereas for MNIST we use a larger value of $\epsilon_{\text{train}}$, following Gowal et al. (2018) and Shi et al. (2021). The values are displayed in Table 3. To schedule $\epsilon$, we set $\epsilon = 0$ for the first epoch of training, and afterwards we smoothly increase $\epsilon$ following Xu et al. (2020). The $\epsilon$ schedule interpolates between $\epsilon = 0$ and $\epsilon = \epsilon_{\text{train}}$ in a smooth manner: for the first $\frac{1}{4}$ of the ramp-up iterations, $\epsilon$ increases according to a degree 4 monomial. After this, $\epsilon$ increases linearly until it reaches a final value of $\epsilon_{\text{train}}$. The linear slope and coefficient of the monomial are chosen so that the derivative of $\epsilon$ is continuous. For CIFAR10 with $\epsilon = 8/255$, we use 159 ramp-up epochs; otherwise, we use 79 ramp-up epochs. For models trained on MNIST, we also use the IBP initialization proposed by (Shi et al., 2021) for all explicit layer weights, which helps prevent IBP bounds from exploding at initialization.

All models besides the DEQ-3 can be trained within a day on a single NVIDIA TitanXp GPU. The DEQ-3 can take up to two days because the hidden layer size is larger (128 output channels), so each iteration of the equilibrium solver is more costly. In all, the training times of the IBP-MonDEQ models vary based on the current model weights but generally are 3-10 times longer than the fully-explicit models. As it is already known that DEQ models tend to compare unfavorably to fully explicit models in terms of runtime (Bai et al., 2019; 2021), improving the runtime discrepancy is deferred to future work.

Finally, we provide additional IBP-MonDEQ-specific implementation details. For IBP-MonDEQ layers, we set $m = 0.1$ in DEQ-3 models and $m = 0.5$ in DEQ-7 models. We find that $m = 0.5$ outperforms $m = 0.1$ for DEQ-7 models despite the reduced expressivity, likely due to optimization difficulties. We set $\gamma = 3$ when training models with the more relaxed LBEN parameterization. For DEQ-7 models, we also find it helpful for optimization to use a warmup schedule on the value of $m$ matching the schedule on $\epsilon$, and set $m_{\text{current}} = m + (0.99 - m)(1 - \frac{\epsilon_{\text{current}}}{\epsilon_{\text{train}}})$ so that the value of $m_{\text{current}}$ interpolates between 0.99 and $m < 0.99$ throughout training. The stopping criterion for the fixed-point solver is $\frac{\|z_{k+1} - z_k\|_2}{\|z_{k+1}\|_2} \leqslant t$, where $t = $1e-5 is the tolerance threshold, and $z_k$ denotes the $k$-th iterate of the fixed-point solving method. For models with the LBEN parameterization, we tie the trainable scaling parameter $\beta$ across channels, so that $\beta$ has the same shape as the convolutional bias parameter.

**Details for Figure 2.** The models trained in Figure 2 (left) are obtained by taking the 7LAYER model and adding additional CONV-128-3x3-1 layers to reach the desired depth. For Figure 2 (right), the bounds are computed for randomly initialized models. We obtain the bounds by attempting to solve (3.3) for DEQ models where $W$ corresponds to a fully-connected layer with output dimension 128. We obtain these bounds on MNIST with $\epsilon = 0.1$ and average the mean of $\bar{z}_k - \underline{z}_k$ for each $k$ over the entire MNIST test set. We use the damped forward-backward algorithm as the equilibrium solver with damping factor $\alpha = 0.1$ (see (A.1)). We note that the standard equilibrium iteration (2.4) converges for both MonDEQ and IBP-MonDEQ models, whereas (3.3) fails to converge for MonDEQ models only.

