# OpenReview forum: "Certified Robustness for Deep Equilibrium Models via Interval Bound Propagation"
_ICLR.cc/2022/Conference — ICLR 2022 Poster_

### Official Review · Reviewer_jEVJ · 2021-10-29

**Correctness:** 3
**Technical Novelty And Significance:** 3
**Empirical Novelty And Significance:** 3
**Recommendation:** 8
**Confidence:** 3

**Main Review:**

Implicit layers such as DEQs are a promising tool in machine learning. Creating DEQ models that admit unique fixed points is an important research question, as is developing deep learning models that are certifiably robust. This work addresses both questions simultaneously, and its theoretical contributions serve as useful tools. On the dow side, only fully connected/convolutional layers are considered in the theoretical framework. There is no obvious theoretical extension of these results to skip connections, transformer architectures, or other complicated deep learning models. As a result, the empirical evaluation is far from what one might be interested in implementing in practice. Still, I think that addressing these problems for fully connected and convolutional layers is an important contribution and a good first step, and the empirical evaluation serves its primary purpose of instantiating the theory presented in this paper.

Please address these questions/comments in your rebuttal. I am willing to upgrade my confidence score if you can address all of these questions.
1. In the statement of theorem 3.1, what does |M| mean? It is mentioned in section 3.2 but I think it would help the reader to move this definition to before it is used for the first time. Is \mathbf{1} a vector of all ones?
2. I am confused by example 3.6. I thought W was required to be symmetric (since I - W should be PD in Proposition 2.1). But this matrix is not symmetric.
3. In the experiments, you mention that you use batch normalisation. Is it true that your theory (utilising 3.6) does not analyse the use of batch normalisation?
4. In the proof of Lemma 3.4, you make reference to \widetilde{W}, which I cannot find a definition for. I think it should be \hat{W}.
5. Can you explain further the last sentence of the proof of Lemma 3.4? I don't understand how (3.8) follows.
6. I can't see where the matrix S comes from in the proof of theorem 3.1. We have E(I - |W|)E = B for B SSD. Writing B = A - ESE we obtain |W| = I-E^{-1} A E^{-1} +S. But how do you know the restrictions on A and S?
7. I do not understand why contribution 1) is separated from 2). Perhaps I missed something subtle in the words. Does "stable" have the same meaning as "guaranteed unique fixed point"? If so, I suggest merging contribution 1) and 2).

Minor points:
- Missing space after period. "...via closed-form functions.These layers are promising ..."
- Winston & Kolter (2020). The arxiv version in your references is now published in NeurIPs.

**Summary Of The Paper:**

The authors propose a deep equilibrium (DEQ) layer that provides certifiable robustness via the interval bound propagation technique. This involves augmenting the original fixed point condition considered in DEQs with two additional fixed point conditions, one for each bound. The main contribution is a theoretical result that says that when parameterised in a certain way, this IBP-DEQ admits a unique fixed point (Theorem 3.1, 3.3), and also that the model provides valid IPB (Proposition 3.2). Motivated by a theoretical result (Proposition 3,7), the authors show empirically that such restrictions imposed by the specific DEQ and parameterisation achieve comparable or improved performance compared with explicit models on MNIST and CIFAR10.

**Summary Of The Review:**

A theoretical paper that develops a new class of certifiably robust deep equilibrium models with guaranteed unique fixed points, a worthy open problem. The model is largely theoretical (i.e. basic networks such as fully connected or convolutional layers), and as such will not achieve state of the art predictive performance. Nevertheless, the empirical evaluation serves its purpose at demonstrating the utility of the model.

---

> ### Author Response · Authors · 2021-11-23
> **Thank you for the review.**
>
> We thank the review for the positive and detailed review which notes that “[the] theoretical contributions serve as useful tools” and “the empirical evaluation serves its primary purpose of instantiating the theory presented in this paper”. Responses to questions below:
>
> -- “In the statement of theorem 3.1, what does |M| mean? … mentioned ... but ... would help the reader to move this definition to before it is used for the first time”
>
> We thank the reader for this suggestion. |M| denotes the entry-wise absolute value of M. We have moved this definition earlier to the statement of Theorem 3.1.
>
> -- “Is \mathbf{1} a vector of all ones?”
>
> \mathbf{1} is indeed a vector of all ones, as defined in the paragraph under eq. 2.3.
>
>  -- “I thought W was required to be symmetric (since I - W should be PD in Proposition 2.1).”
>
> We note that under Proposition 2.1, PD is defined to mean that u^\top A u > 0 for all nonzero vectors u and PD matrix A. Note that under this definition, the matrix A does not have to be symmetric. The revision clarifies this distinction more prominently.
>
> -- “Is it true that your theory (utilising 3.6) does not analyse the use of batch normalisation?”
>
> Our theory does indeed not analyze the use of batch normalization. However, nothing in the experiments contradicts the theory, because all BatchNorm layers are *outside* of the equilibrium-solving layers, and are instead located in the explicitly-defined parts of the network.
>
> -- “you make reference to \widetilde{W}, which I cannot find a definition for. I think it should be \hat{W}.”
>
> We thank the reviewer for catching this mistake -- it should indeed be \hat{W}.
>
> -- “Can you explain further the last sentence of the proof of Lemma 3.4? I don't understand how (3.8) follows.”
>
> To see the left-to-right implication of (3.8), we use the fact that the set of eigenvalues of $\widehat{W} + \widehat{W}^\top$ contains those of $|W| + |W|^\top$ as a subset. For the right-to-left implication, we note that $I - |W| \succ 0$ implies that $\lambda_{\max}((|W| +|W|^\top)/2) < 1$. Next, we can observe that $\lambda_{\max}(|W| + |W|^\top) = \lambda_{\max}(\widehat{W} + \widehat{W}^\top)$ because eigenvalues of $\widehat{W}+ \widehat{W}^\top$ are either eigenvalues of $W + W^\top$ or eigenvalues of $|W| + |W|^\top$. Now we can use the fact that $\lambda_{\max}(W + W^\top) \le \lambda_{\max}(|W| + |W|^\top)$, which would give us $\lambda_{\max}((\widehat{W}+ \widehat{W}^\top)/2) < 1$, or $I - \widehat{W}\succ 0$.
>
> Finally, to see that $\lambda_{\max}(W + W^\top) \le \lambda_{\max}(|W| + |W|^\top)$, let $v$ be the eigenvector corresponding to the maximum eigenvalue of $W + W^\top$. The key observation is that $v^\top (W + W^\top ) v \le |v|^\top (|W| + |W|^\top) |v|$ because terms in the expansion of the first quantity will be replaced by positive versions with the same magnitude in the r.h.s.
>
> We have included these details in the revision.
>
> -- “I can't see where the matrix S comes from in the proof of theorem 3.1”
>
> We can first write $(|W| + |W|^\top)/2 = I - E^{-1} A E^{-1}$. Now note that $|W| = (|W| + |W|^\top)/2 + (|W| - |W|^\top)/2$, and that the second term is a skew-symmetric matrix. The first term can be replaced with $I - E^{-1} A E^{-1}$, giving an end result of the form $|W| = I - E^{-1} A E^{-1} + S$.
>
> -- “I do not understand why contribution 1) is separated from 2).”
>
> Contributions 1) and 2) are separate because they reflect two different interpretations of our result. The first, as noted in contribution 1), is that our result demonstrates how to make DEQs certifiably robust, and is therefore of interest to researchers who study DEQs. Second, our result is interesting in its own right for researchers who care about IBP, because IBP has not worked very well for deep networks due to stability issues. One way to interpret our result is that we get IBP to work for a class of infinitely deep networks (since MonDEQs can be interpreted as such). This perspective is reflected in contribution 2).

---

### Official Review · Reviewer_wCmR · 2021-11-02

**Correctness:** 3
**Technical Novelty And Significance:** 3
**Empirical Novelty And Significance:** 3
**Recommendation:** 6
**Confidence:** 3

**Main Review:**

The authors present a non-trivial extension of IBP to DEQ models. A fixed-point version the standard IBP technique is presented, which involves imposing additional constraints on the DEQ weights in order to guarantee that a unique fixed-point solution exists. Two different settings are considered: MonDEQ, and the less restrictive LBEN parametrization. It is shown that the IBP-friendly version of the LBEN parametrization maintains enough expressive power to represent explicit feedforward networks.

The theoretical analysis seems solid and quite comprehensive, and I believe that the experimental results are interesting on their own, as they present the first results for robustly trained DEQ models. However, I think the quality of the paper would benefit from addressing the following points:
- Figure 2 shows that (UB - LB) grows with the iterations of the fixed-point solver (until convergence). I believe this implies that the output of the fixed-point solver is a valid bound on the DEQ layer activations only if it is run until convergence. Is this indeed the case? If so, do the authors take this into account (that is, is it always run to convergence in the experiments)?
- On top of the training experiments, it would be interesting to assess the tightness of the IBP-MonDEQ bounds (to equation (2.1)) of a pre-trained network (both trained via vanilla SGD, and via IBP-MonDEQ robust training). For instance, they could be plotted against the bounds obtained via a PGD attack.
- The experimental results show that IBP-MonDEQ networks are competitive with IBP-trained explicit networks. However, other robust training methods for explicit networks exist. Would IBP-MonDEQ training be still competitive against CROWN-IBP, for instance?
- Appendix B reports that the runtime overhead is quite large: a factor 3-10 compared to explicit networks. This seems to be quite a large overhead, given that the empirical gains seem to be limited. Do the authors believe that these results call for the use of DEQ models for adversarial robustness? Or rather that explicit models are still more advantageous?

Minor comments:
- Isn't (2.2) generally defining lower/upper bounds rather than IBP specifically?

**Summary Of The Paper:**

The paper presents IBP-MonDEQ, a modification of monotone deep equilibrium layers (MonDEQ) that allows for the computation of lower and upper bounds on its output via (a fixed-point version of) interval bound propagation. As a result, the authors can train a certifiably robust DEQ model, whose performance is shown to be competitive with an explicit model trained via IBP.

**Summary Of The Review:**

IBP-MonDEQ is an interesting and non-trivial extension of IBP to deep equilibrium layers that allows for relatively effective robust training of DEQ networks. The theoretical analysis seems comprehensive, and the experiments show that the proposed algorithm works in practice. I believe the paper would further improve by extending the experimental section and addressing its limitations (large runtime overhead).

---

> ### Author Response · Authors · 2021-11-23
> **Thank you for the review.**
>
> We thank the reviewer for the insightful review and interesting questions and for pointing out that the “IBP-MonDEQ is an interesting and non-trivial extension of IBP” and that the “theoretical analysis seems comprehensive.” We will respond to specific points below.
>
> -- “... output of the fixed-point solver is a valid bound on the DEQ layer activations only if it is run until convergence. Is this indeed the case? If so, do the authors take this into account (that is, is it always run to convergence in the experiments)?”
>
> It is indeed true that the fixed-point solver only outputs a valid interval bound when run to convergence. We do run to convergence in the experiments (so that the relative change in the iterates of the solver has norm bound 1e-5).
>
> -- “would be interesting to assess the tightness of the IBP-MonDEQ bounds (to equation (2.1)) of a pre-trained network (both trained via vanilla SGD, and via IBP-MonDEQ robust training). For instance, they could be plotted against the bounds obtained via a PGD attack.”
>
> We thank the reviewer for the suggestion. Evaluating robustness against PGD attacks is definitely an interesting experiment for future revisions of this work, and understanding robustness of DEQs to PGD attacks is an interesting and more broad future direction.
>
> -- “The experimental results show that IBP-MonDEQ networks are competitive with IBP-trained explicit networks. However, other robust training methods for explicit networks exist.”
>
> Our comparison indeed focused on IBP-trained explicit networks because they are the direct counterpart of our method for explicit networks and can therefore serve to contextualize the performance of the IBP-MonDEQ models. However, we note that our goal is to develop algorithms for certifying robustness *of DEQs*, not to be competitive against all certified robustness algorithms for all architectures. Given this goal, we feel that comparing against certified robustness methods which are not vanilla IBP is not necessary for achieving the main goals of this work.
>
> -- “... runtime overhead is quite large ... Do the authors believe that these results call for the use of DEQ models for adversarial robustness? Or rather that explicit models are still more advantageous?”
>
> See the above response to the previous point -- we do not intend to call for the use of DEQ models over explicit models for adversarial robustness. Rather this work, and many other works on DEQs in general, are more exploratory in nature. The main focus is on understanding whether it is even possible to make DEQs certifiably robust to $\ell_{\infty}$ attacks, rather than immediately propose practical algorithms using DEQs.

---

> > ### Comment · Reviewer_wCmR · 2021-11-29
> > **Final thoughts**
> >
> > I thank the authors for their response.
> >
> > After reading the rebuttal and the other reviews, I decided to retain my score of 6.
> > As the authors said, the work is "exploratory in nature" (the bounds are valid only at fixed point convergence, runtime overhead compared to explicit networks, results below the state-of-the-art).
> > However, I believe the approach is still interesting, as it provides the first robust training method for DEQs.
> > Given the provided empirical results, I would personally draw a negative albeit interesting conclusion: at present, the overhead associated to DEQs does not pay off in terms of robust accuracy (the gains are not commensurate with the overhead).

---

### Official Review · Reviewer_KViU · 2021-11-02

**Correctness:** 4
**Technical Novelty And Significance:** 3
**Empirical Novelty And Significance:** 3
**Recommendation:** 5
**Confidence:** 3

**Main Review:**

Strengths:
* This paper derives parameterizations for DEQ such that the fixed-point solution of the DEQ augmented with IBP bounds exists and is unique.
* The paper theoretically show that a single IBP-MonDEQ layer can express all explicit feedforward networks.
* The paper empirically demonstrates the certified defense of IBP-MonDEQ.

Weaknesses:
* The empirical results are limited which do not show significant improvement over traditional CNN (As shown in the paper, on CIFAR-10, the improvement is no more than 0.5 percentage points compared to traditional CNN used in this paper).
* Baselines results are too weak -- the performance of the baselines is much worse than state-of-the-art results (e.g.,  Shi et al., 2021). For example, on CIFAR-10 eps=8/255, Shi et al., 2021 shows certified error 65.58 ± 0.32 (or 67.01 ± 0.29 partly without their improvements), but the baseline in this paper has higher errors 69.51 ± 0.46.

**Summary Of The Paper:**

This paper considers the certified adversarial robustness of Deep Equilibrium Models (DEQ) and derives the Interval Bound Propagation (IBP) on DEQ for training certifiably robust models, namely IBP-MonDEQ.

**Summary Of The Review:**

This paper is interesting and can be meaningful for further research as it derives the IBP computation on DEQ for certified defense. But the empirical results are limited so far.

=========Post-rebuttal updates=========

Thanks for the response from the authors. The authors have improved their experimental results where the baselines match better with [Shi et al.’21] (currently this update is not visible to the public).

This paper does present interesting findings in terms of verifying DEQ and can be potential interesting for the area of DEQ, and this is the first work for verifying DEQ which is different from explicit networks.

But the contribution is kind of insufficient. The paper fails to demonstrate the benefit of using DEQ itself. If it is unclear why we need DEQ, it will be minor to study IBP bounds for DEQ, especially  IBP-MonDEQ fails to really outperform baselines with explicit models (very marginal gap). I think it may help if the authors can introduce some scenaraios in the experiments that are particularly suitable for applying DEQ, to demonstrate that DEQ is really needed.

---

> ### Author Response · Authors · 2021-11-23
> **Thank you for the review.**
>
> We thank the reviewer for the insightful review. Our responses to the reviewer’s points are as follows:
>
> -- “empirical results are limited which do not show significant improvement over traditional CNN”
>
> We’d like to respectfully point out that the goal of this paper is not to argue that IBP-MonDEQs should be the go-to architecture for $\ell_{\infty}$ certified robustness. Rather, our goal is to demonstrate how to certify $\ell_{\infty}$ robustness *for DEQs*, and the experiments are more for sanity-checking the model and verifying that their performance is reasonable. From this vantage point, we’d argue that the empirical comparison accomplishes its desired purpose.
>
> -- “performance of the baselines is much worse than state-of-the-art results (e.g., Shi et al., 2021)”
>
> We thank the reviewer for pointing out this discrepancy. The baseline and IBP-MonDEQ were run with hyperparameters such as number of epochs, epsilon schedule, etc set to the same values. Regardless, we are running experiments to address this discrepancy. When the experiments are finished, we will follow up on the results in a future comment.

---

### Official Review · Reviewer_7ZJs · 2021-11-04

**Correctness:** 2
**Technical Novelty And Significance:** 3
**Empirical Novelty And Significance:** 3
**Recommendation:** 3
**Confidence:** 5

**Main Review:**

Pros:
1. Standard training schemes and architectures cannot guarantee accuracy and certified robustness at the same time. The work focuses on addressing this issue and therefore the problem considered here is an important and challenging one.

2. The construction of IBP-MonDEQ is novel and is among the first attempts to create certified implicit networks. While the interval analysis is relatively simple, the authors provide non-trivial theoretical analysis and guarantees.

Cons:
1. Some of the details about theoretical formalism and empirical evaluation was not clear to me (see my comments below).

2. The practical relevance of the certified robustness obtained on the more challenging CIFAR10 dataset presented here is not clear as the results do not advance the state-of-the-art. For example, the best-certified robustness for CIFAR10 for epsilon=2/255 achieved here is 51% with the corresponding standard accuracy as 64%. This is significantly worse than that obtained by the state-of-the-art which is 60.4% robustness and 78.4% accuracy as reported by COLT (https://openreview.net/pdf?id=SJxSDxrKDr). Similarly for epsilon=8/255, the best-certified robustness and accuracy achieved here are 30% and 44% respectively. These are again worse than the state-of-the-art bounds of 35% robustness and 50% accuracy from L_oo nets (https://arxiv.org/pdf/2102.05363.pdf).

I have a few other questions:

1. Since the interval abstraction is a complete lattice. Therefore, fixed points exist for any monotone (wrt interval inclusion) neural network. Is the class of networks that you identify in Theorem 3.1 and 3.3 a subset of those provided by the classical Knaster-Tarski theorem (https://en.wikipedia.org/wiki/Knaster%E2%80%93Tarski_theorem)?

2. Is it possible to define a class of DEQs that have multiple fixed points?

3. Can one equivalently identify a class of fixed-point obtaining implicit networks for other analysis types? e.g., CROWN, DeepPoly, or Zonotopes?

4. The authors should consider providing an intuitive meaning of symbols (e.g., M, D) used in equations (3.4) and (3.7). The text in the evaluation says that M is a learnable parameter which makes things clearer but it comes too late. An example showing instances of W that satisfy these equations and those that do not would also help in improving the readability of the paper.

5. Is it possible to train IBP-MonDEQs for perturbations that cannot be exactly captured by intervals?

6. Is the certified robustness in Table 2 for explicit networks computed using IBP analysis? If yes, will the numbers improve with a complete verifier or with a more precise analysis like Crown or DeepPoly? This is important as it seems that the IBP-MonDEQ cannot be analyzed with anything else besides the IBP analysis while the explicit networks support other analyses.

7. What are the implications of computing a post fixed-point by replacing the equality constraint in eq. (3.6) with interval inclusion, i.e., compute a fixed-point such that the interval of RHS is included inside that for LHS?

**Summary Of The Paper:**

The paper presents a new class of neural networks called IBP-MonDEQs that are an extension of the recently introduced implicit networks MonDEQs. The authors identify a class of weight matrices of the network that ensures that fixed point of the implicit layers exist with respect to the interval analysis and is unique. The construction of IBP_MonDEQ is motivated by the goal of obtaining networks that can be certified to be robust. The authors then train such networks comparing against explicit networks of the same architecture constructed using the certified IBP training. The results show that IBP-MonDEQs can obtain better certified robustness than explicit networks on the MNIST and CIFAR10 dataset.

**Summary Of The Review:**

The paper makes solid theoretical and empirical contributions for obtaining new implicit architectures with certified guarantees on their robustness. However, I am not sure whether the  proposed method advances the state-of-the-art in certified robustness unlike the L_oo nets. Further, many key details were not clear to me including comparison with the classical Knaster-Tarski theorem and whether the explicit networks are certified with complete verifiers or IBP.

---

> ### Author Response · Authors · 2021-11-23
> **Thank you for the review.**
>
> We thank the reviewer for the detailed and insightful review. Our response is below:
>
> -- “practical relevance … is not clear as the results do not advance the state-of-the-art”
>
> We respectfully disagree that our approach should be compared against SOTA approaches for certified robustness. The comparison against IBP for explicit networks is intended to contextualize the performance of IBP-MonDEQs against similar algorithms for traditional neural net architectures, not to argue that the IBP-MonDEQ is competitive against all certified robustness methods.
>
> Indeed, because the understanding of DEQ models is still nascent, many papers on DEQs are still exploratory in nature and do not immediately lead to algorithms which are competitive against SOTA. As our paper is written from the perspective of advancing DEQs, our belief is that the more appropriate evaluation metric should be against other certified robustness methods for DEQs, *not for all possible certified robustness methods.* From this perspective, our paper succeeds because prior to our work, it was unknown how to train \ell_{\infty}-certifiably-robust DEQs.
>
> -- “Since the interval abstraction is a complete lattice … Is the class of networks that you identify in Theorem 3.1 and 3.3 a subset of those provided by the classical Knaster-Tarski theorem” “comparison with the classical Knaster-Tarski theorem”
>
> We thank the reviewer for this suggestion. Actually, the Knaster-Tarski theorem does not apply here since the intervals *do not* form a complete lattice, because they are unbounded, i.e, the interval lower bounds could be arbitrarily small or the upper bounds could be arbitrarily large. Complete lattices are bounded; for example, the set of positive integers does not form a complete lattice under the standard ordering (1 < 2 < 3 < …). (We note that the function f(x) = x + 1 is monotone w.r.t. this ordering, but trying to apply Knaster-Tarski would give an erroneous conclusion that f has a fixed point.)
>
> -- “Is it possible to define a class of DEQs that have multiple fixed points?”
>
> It is likely possible; for example, vanilla DEQs without any constraints parameterization may have multiple fixed points. However, in such a case, the output of the DEQ is technically not mathematically well-defined and in practice could vary based on the randomness in the equilibrium solver. For certified robustness, it’s therefore important that the DEQ is guaranteed to have a *single* fixed point.
>
> -- “Can one equivalently identify a class of fixed-point ... networks for other analysis types? e.g., CROWN, DeepPoly, or Zonotopes?” “Is it possible to train IBP-MonDEQs for perturbations that cannot be exactly captured by intervals?”
>
> We thank the reviewer for these suggestions -- these are interesting research directions that could build off of our current work.
>
> -- “authors should consider providing an intuitive meaning of symbols (e.g., M, D) used in equations (3.4) and (3.7).”
>
> M can be thought of as an unconstrained matrix (and, as pointed out by the reviewer, is also treated as an unconstrained parameter in the actual implementation). D is then a diagonal scaling (which depends on M) which guarantees that the matrix W = D^{-½} M D^{-½} is a valid weight matrix for the IBP-MonDEQ. We have incorporated this discussion into the revision.
>
> -- “Is the certified robustness in Table 2 for explicit networks computed using IBP analysis? If yes, will the numbers improve with a complete verifier or with a more precise analysis like Crown or DeepPoly?”
>
> Yes, results are certified via IBP. We suspect that the numbers would improve with other verification methods, but this is orthogonal to our central goal (see our first response in this rebuttal.)
>
> --  “IBP-MonDEQ cannot be analyzed with anything else besides the IBP analysis”
>
> We completely agree -- certified robustness methods have all been designed for traditional explicit networks, and do not immediately work for DEQs, which makes certifying robustness of DEQs an interesting and important question in its own right.
>
> -- “implications of computing a post fixed-point by replacing the equality constraint in eq. (3.6) with interval inclusion”
>
> We thank the reviewer for this interesting question. One implication is that the true fixed point of (3.6) would have to be included inside this inclusion-based fixed point from the reviewer’s question. To see this, let $f$ denote the function for the fixed-point equation, and recall that interval inclusion defines an ordering $\le$ on the interval bounds such that $f$ is monotone w.r.t. this ordering (as pointed out by the reviewer previously.) Then the reviewer is asking what happens if we have $x$ such that $f(x) <= x$. By monotonicity of $f$, this also means $f(f(x)) \le f(x)$, and so on, so $x \ge f(x) \ge f(f(x)) \ge …$. Then the limit of applying $f$ infinite times would give the true fixed point of $f$, which would be included in the original interval corresponding to $x$.

---

> > ### Comment · Reviewer_7ZJs · 2021-11-28
> > **Post Response Impressions**
> >
> > Dear Authors,
> >
> > Thanks for the response, it answered some of my questions. However, I have still two main concerns:
> >
> > 1. I am not still not convinced about the practical relevance of the work. The IBP method is already the cheapest method for training provably robust networks. The IBP on proposed networks is already significantly (3-10)x slower than explicit networks. It is not clear where the improvements are going to come from in the future as methods like CROWN-IBP or Zonotopes will be even more expensive. In my opinion, there might be some other practical application domains beyond vision with the usual MNIST and CIFAR10 where the considered training methods can provide better results and I encourage the authors to investigate that.
> >
> > 2. The claim made in the response that the intervals do not form a complete lattice is **incorrect**. The set of positive integers can be easily extended with {-oo,+oo} to make the interval lattice defined on them to be complete (this also works for the reals). See Example 2.6 and Fig. 2.3 here (https://hal.sorbonne-universite.fr/hal-01657536/document). Further, in the setting considered in the paper, the authors do not even need to worry about extending the set with infinities as all the L_oo balls are bounded. Also, see the original work of Tarski (https://msp.org/pjm/1955/5-2/pjm-v5-n2-p11-p.pdf) which applies to Intervals. Based on this, I am not sure whether the work presented here does something new or represents a limited subset of the classical work from the 1950s. I encourage the authors to explore how their work differs from this.
> >
> > In light of these issues, I have decided to reduce my grades.

---

> > > ### Author Response · Authors · 2021-11-30
> > > **Response**
> > >
> > > We thank the reviewer for reading and responding to our rebuttal. Responses to the reviewer's points:
> > >
> > > 1. “still not convinced about the practical relevance of the work”
> > >
> > > The reviewer is still evaluating our work in comparison to certified robustness methods for explicit models. However, as discussed in the original rebuttal, we feel that the most appropriate way to evaluate our work is strictly within the context of studying and advancing DEQ models. Within this context, our work takes an important step by demonstrating how to certify $\ell_{\infty}$ robustness for DEQs, which was previously unclear.
> > >
> > > 2. “The claim made in the response that the intervals do not form a complete lattice is incorrect.”
> > >
> > > We respectfully disagree with the reviewer’s point here -- we stand by our original point that the Knaster-Tarski Theorem does not apply in a non-trivial way. First:
> > >
> > > -- “in the setting considered in the paper, the authors do not even need to worry about extending the set with infinities as all the L_oo balls are bounded”
> > >
> > > In our setting, when we try to find the equilibrium point $\begin{bmatrix} \bar{z} \\ \underline{z} \end{bmatrix} = f(\begin{bmatrix} \bar{z} \\ \underline{z} \end{bmatrix})$, the input domain of f is *unbounded* because entries of $\bar{z}$ can be arbitrarily large and entries of $\underline{z}$ can be arbitrarily small. Thus, we *do* have to extend to the setting with infinities to apply the Knaster-Tarski theorem to our paper.
> > >
> > > -- “set of positive integers can be easily extended with {-oo,+oo} to make the interval lattice defined on them to be complete (this also works for the reals)”
> > >
> > > Even in the case where infinities are included in the lattice, we don’t foresee how the Knaster-Tarski theorem would guarantee the existence of *nontrivial* equilibria solutions. Let’s consider the example from the original rebuttal of positive integers $1 < 2 < 3 < … < \infty$, where there is a lattice element $\infty$ making the lattice complete. Consider the case of the monotone function $f$ satisfying $f(x) = x+ 1$ for finite $x$, and $f(\infty) = \infty)$. By Knaster-Tarski there is an equilibrium point, but clearly the only equilibrium point here is the “trivial” equilibrium $\infty$. Thus, the conclusion of Knaster-Tarski isn’t meaningful in this simplified example. This same pathology could arise when trying to apply Knaster-Tarski to the lattice on intervals (which must contain $\infty$, as argued above).

---

> > > > ### Comment · Reviewer_7ZJs · 2021-12-06
> > > > **Keeping my rating**
> > > >
> > > > I thank the authors for their detailed response, however, it did not address my concerns.
> > > >
> > > > Besides not having state-of-the-art results, there are also issues in comparison with explicit IBP models. One of the main claims here is that the proposed approach can achieve competitive performance wrt explicit IBP trained models. As mentioned by the authors in their response, the certified bounds are currently computed using the weaker IBP analysis for explicit networks and the numbers will improve with more precise analysis. If one looks at the state-of-the-art IBP networks presented in https://arxiv.org/pdf/1810.12715.pdf, the certification is performed with complete verifiers based on MILP solvers. If one can devote 1-2 days to training IBP Mon-DEQ networks, then why cannot we have more precise certification methods for explicit networks? Overall, I believe that the experimental setting currently seems to favor the IBP MonDEQ networks.
> > > >
> > > > There have been new, expensive architectures proposed recently in https://arxiv.org/abs/2105.11363 that have achieved state-of-the-art performance on smaller datasets/networks (which is why I encouraged the authors to look into other datasets in my previous comment). Therefore I am not convinced about the argument that one should not expect state-of-the-art performance with IBP based MonDEQ proposed here.
> > > >
> > > > One can prove by induction that f is bounded. For the base case, the input region is bounded. Now assume after k iterations, the inputs to ReLU(Wz+v(x)) are bounded. Since for bounded input, both affine transformation (of the type considered in NNs) and ReLU compute bounded output, induction holds and oo are not needed. This is actually the reason why the Anderson acceleration used in this paper computes non-oo weights. There is nothing special about the Anderson acceleration that will allow it to discover non-trivial fixpoints when they do not exist. I ran the Anderson acceleration code from https://en.wikipedia.org/wiki/Anderson_acceleration on the function f(x)=x+1 and it gave me oo as the fixed point. Therefore, I am not sure about the relevance of this example given by the authors.

---

> > > > > ### Author Response · Authors · 2021-12-07
> > > > > **Response**
> > > > >
> > > > > We thank the reviewer for the discussion.
> > > > >
> > > > > -- “issues in comparison with explicit IBP models …  If one can devote 1-2 days to training IBP Mon-DEQ networks, then why cannot we have more precise certification methods for explicit networks?”
> > > > >
> > > > > If our goal were to argue that IBP-MonDEQ models should replace explicit models for certified robustness, then maybe comparison against exact certification methods could be reasonable; however, *this is not our main goal.* Our main goal is to demonstrate how to certify l_infty robustness for DEQs, which was previously unclear, and the main goal of the experiments is to sanity-check the proposed IBP-MonDEQ models by comparison against similarly-trained explicit networks. As DEQs tend to suffer from the additional runtime overhead of equilibrium solving, the slower runtime of the IBP-MonDEQ v.s. explicit network is hard to avoid. Eliminating this runtime discrepancy is orthogonal to the main purpose of our experiments.
> > > > >
> > > > >  -- “new, expensive architectures ... that have achieved state-of-the-art performance on smaller datasets/networks (which is why I encouraged the authors to look into other datasets in my previous comment).”
> > > > >
> > > > > We thank the reviewer for the suggestion, which we will explore in future work. We’d like to respectfully point out again that achieving SOTA on some dataset doesn’t seem necessary to accomplish our main goal, which is to study certified robustness *within the context of DEQ models*.
> > > > >
> > > > > -- “One can prove by induction that f is bounded.”
> > > > >
> > > > > If the reviewer is arguing that our results follow trivially from the Knaster-Tarski theorem, we still respectfully disagree and stand by our original responses. We’d like to point out below that the reviewer’s logic for proving that f is bounded does not seem correct.
> > > > >
> > > > > “Bounded” in this context refers to the property that $f$ maps a bounded lattice $L$ to the *same* lattice $L$. It’s non-trivial to prove that this property holds -- the only way we know how to prove it is to go through the exact arguments in our paper. Furthermore, if $f$ is “bounded”,  it’s not clear why the induction aspect of the reviewer’s argument is even needed. The reviewer appears to have assumed some form of boundedness in their argument (“both affine transformation (of the type considered in NNs) and ReLU compute bounded output”), which makes us suspect that their proof is incorrect.

---

### Decision · Program_Chairs · 2022-01-20

**Decision:**

Accept (Poster)

**Comment:**

Note: This meta review is written by the SAC, but it's synced with the AC.

Summary (adopted from Reviewer wCmR): This paper presents a modification of monotone deep equilibrium layers that allows to compute the bounds on the output via the IBP algorithm. This also allows to train a certifiably robust DEQ model with a competitive performance.

Initial reviews were mixed, but post rebuttal the opinions generally improved. Reviewer wCmR intend to increase their score slightly (6 to 7) and Reviewer KViU also mentioned that their opinion improved. Reviewer 7ZJs maintained their score and, during discussion phase, made many arguments against acceptance. One of those was about Tarski's theorem which was deemed not so important by the AC and also KViU. Another concern was about experimental results to which KViU agreed, and this remains the main concern for now.

Most reviewers agree that the work is interesting and is a good step, but then utility of the new modification and significance of the results remains a question. It is likely that the work may be useful in the future, and as there is an overall increase in the opinion, I believe that it is okay to accept the paper.

I encourage the authors to take the comments of the reviewers into account, and clearly mention the issues raised in the paper.

SAC